# Infection Prevention Mask Consisting of Nanofiber Filter and Habutae Silk Fabrics

**DOI:** 10.3390/ma14237391

**Published:** 2021-12-02

**Authors:** Masayo Suekawa, Yuya Hashizume, Shuichi Tanoue, Hideyuki Uematsu, Yoshihiro Yamashita

**Affiliations:** 1Faculty of Education, Humanities and Social Sciences, University of Fukui, Bunkyo 3-9-1, Fukui 910-8507, Japan; suekawa@u-fukui.ac.jp; 2Industrial Innovation Engineering, Graduate School of Engineering, University of Fukui, Bunkyo 3-9-1, Fukui 910-8507, Japan; yuya.zume@gmail.com (Y.H.); tanoue@u-fukui.ac.jp (S.T.); uematsu@matse.u-fukui.ac.jp (H.U.); 3Research Center for Fibers and Materials, University of Fukui, Bunkyo 3-9-1, Fukui 910-8507, Japan

**Keywords:** nanofiber, silk fabric, habutae, virus, SARS-CoV-2, mask, hypoallergenic, air permeability

## Abstract

To reduce skin irritation and allergic symptoms caused by long-term mask use, we produced a mask with a filter effect by laminating nanofibers on habutae silk fabric, a specialty of Japan’s Fukui Prefecture, using the electrospinning method. We investigated the filter characteristics of silk fabrics with different weave structures (habutae, flat crepe, and twill). We found that woven fabrics alone could not sufficiently block particles finer than 1 μm, even when the fabric layers were overlapped. Therefore, we had a nanofiber filter layer fabricated on the surface of habutae fabric by the electrospinning method at a weight of 1 g/m^2^. The nanofibers removed more than 94% of 0.3 μm-particles, which are similar to the size of virus particles. However, the nanofiber layer was so dense that it caused an increase in pressure drop, so we made the nanofiber layer thinner and fabricated the filter on the surface of the habutae fabric at 0.5 g/m^2^. A three-dimensional mask consisting of two woven fabrics, one with a nanofiber layer on the inside and the other with a normal woven fabric without a nanofiber layer on the outside, was fabricated and tested on 95 subjects. The subjects reported that the nanofiber habutae masks were more comfortable than nonwoven masks. Moreover, the silk woven masks did not cause allergic symptoms such as skin irritation.

## 1. Introduction

While SARS-CoV-2, which has been widespread since December 2019, is expected to end with the development of vaccines and drugs, concerns remain about the emergence of new mutant strains. The global shortage of nonwoven masks to prevent infection continued from the end of January to around May 2020, for this reason, many cloth masks became available. Cloth masks are not suitable for use in medical settings with high virus exposure, but have been reported to be beneficial for use by the general public in low-risk settings where there are no alternatives to nonwoven masks [1]. While nonwoven masks perform well, they also cause skin irritation and other symptoms. Nonwoven masks were reported to cause changes in skin temperature, redness, and TEWL (transepidermal water loss) in the short term, and changes in skin elasticity, pores, and acne in the long term [2]. In addition, amongst the 157 staff who wore FFP3 masks, redness of the nasal area was most frequently reported impact, with 8% reporting facial blisters [3]. Wearing a mask can also cause mental discomfort. Liu et al., reported that subjects felt hotter and more humid while wearing a mask, and that their discomfort increased significantly while wearing the mask, while at the same time their mean skin temperature and heart rate increased, blood oxygen saturation levels decreased, and health and comfort levels ultimately decreased [4]. To prevent the entry of viruses between the edges of a mask and the face, the gap between each edge and the face can be reduced by tightening the mask when it is worn, but the negative effect of tightening on the skin has also been pointed out. Peko et al., reported that common nonwoven masks were less irritating to the face than KN95 masks [5]. The reason for this is that nonwoven masks apply less force to the face and face temperature returns to its basic level quickly. It is also reported that there is no difference between cloth masks and N95 masks in terms of the diffusion of droplets from the mouth to pass through a mask [6].

Silk is a natural fiber and is bio-degradable [7,8] and silk fabrics are known to be less irritating to the skin and less likely to cause allergies. As a result, we focused on silk cloth masks to solve the lack of masks and skin problems caused by wearing nonwoven masks. We have developed masks made of habutae silk, a specialty of Fukui Prefecture, Japan. There have been several reports on the safety of silk to the skin [9,10,11]. It has also been reported that traditional cotton and silk fabrics improve atopic dermatitis symptoms and decrease their severity. In the present study, 95 people were surveyed to determine whether or not they developed skin irritation and allergic symptoms during long-term use of a habutae woven mask. It is well known that the filtering performance of cloth masks is inferior to that of nonwoven masks and N95 masks [12]. However, the relationships between the weave structures of silk fabrics and their filter performance when they are laminated have not been clarified. Konda et al., investigated the particle trapping properties of various fabrics, but it was not a systematic study, and the relationship between the weave structure and the particle trapping rate of silk fabrics had not been clarified [13]. Furthermore, the relationship between the trapping rate and the pressure drops when they are laminated was also unclear. In this study, we tried to clarify the effects of the weaving structure and the number of layers of silk fabric on the particle collection rate.

We also investigated the usefulness of nanofiber layers fabricated on the surface of the fabric by electrospinning to improve filter performance. It is well known that the filtering performance of cloth masks is inferior to that of nonwoven masks and N95 masks [12]. However, the relationships between the weave structures of silk fabrics and their filter performance when they are laminated have not been clarified. We also investigated the usefulness of nanofiber layers fabricated on the surface of the fabric by electrospinning to improve filter performance. The size of viral droplets is micrometers, but one virus is 100 nm in size. We tried to improve the filter performance by spinning nanofibers into the fabric. There have been many reports on the formation of nanofiber filter layers on nonwoven fabrics by electrospinning [14,15,16], but there are few reports about laminating nanofibers on textiles. If nanofibers can be directly laminated onto textiles, it will be possible to use them not only as filters but also as apparel materials with moisture permeability and water repellency. In most cases, PVDF and other polymers are dissolved in organic solvents such as DMF and then spun. However, residual solvents are of concern and their use as masks is undesirable. Therefore, we tried to make an emulsion of water-resistant ethylene-vinyl alcohol copolymer (EVOH)/polyurethane (PU) in water/alcohol and electrospinning it for use in filters. EVOH fiber is not recyclable, but is thermal and chemically stable, and absorbs a small amount of water [17,18]. We also used a large electrospinning machine for mass production. In this way, we verified the possibility of practical application of a highly functional and safe cloth mask by laminating a nanofiber filter layer on a low-allergy habutae fabric.

## 2. Experiment

### 2.1. Silk Fabrics

Typical weaving structures for silk fabrics include plain weaving, twill and satin. In addition, the most common plain weaves are habutae, crepe, flat crepe, taffeta and chiffon. Figure 1 shows electron micrographs of typical silk fabrics. In this study, habutae, twill and flat crepe were considered to be facial mask candidates. Chiffon was excluded because of its large gaps. Silk fabrics from Arai Silk Ltd. (Fukui Prefecture, Japan) were used.

### 2.2. Filter Performance Evaluation

The filter performance of each silk fabric was measured using a filter performance-evaluation tester (DFT-4, Tokyo Direc Co., Ltd., Tokyo, Japan) to measure the collection efficiency and pressure drop of medium- to high-performance filter materials based on JIS B-9908. The air flow rate was 32 L/min, the air velocity was 5.3 cm/s, and the filtration area was 100 cm^2^. JIS 11−type powder (Kanto loam soil, The Association of Powder Process Industry and Engineering, Kyoto, Japan) was used as the fine particles for measurement.

Pressure drop and particle rejection were measured at N = 3. All of the data in the middle of the three measurements were used for the figure. Variability between each measurement were all within ±15%.

### 2.3. Fabrication of Nanofiber Filter Membrane

An emulsion consisting of ethylene-vinyl alcohol/polyurethane (Nihon Cima Co., Ltd., Goka-Machi, Ibaraki, Japan) [19] was used to fabricate the nanofiber filter layer on the surface of habutae fabric. The electrospinning equipment was NS-Lab (Elmarco Co., Ltd., Liberec, Czech Republic), with an applied voltage of 70 kV, a distance of 100 mm between the fabric and the nozzle wire, an ambient temperature of 25 °C and a humidity of 45%. The thickness of the nanofiber layer was evaluated by weight per unit area. At a feed rate of 50 mm/min for habutae fabric, the unit weight was 1 g/m^2^; and at a feed rate of 200 mm/min, it was 0.5 g/m^2^. The 0.5 g/m^2^ habutae fabric was used to test the mask. The fiber diameters and pore size of the nanofibers were also measured in three images.

### 2.4. Sewing of the Masks

The box-type three-dimensional mask had a structure that allows space around the mouth and that follows the lines of the face (as shown in Figure 2) to enhance adhesiveness. Three sizes, L, M, and S, were made by Lacorme Co. Ltd., Katsuyama, Fukui, Japan. Each mask was made of two pieces of habutae fabric sewn together. On the inner side (against the mouth) was sewn habutae fabric with a filter layer (0.5 g/m^2^) made by spinning nanofibers, and on the outer side was placed an untreated habutae fabric. For the elastic loops that go around the ears, we selected a material that would not cause ear pain.

### 2.5. Washing Durability Test

The masks were washed according to the JIS L 1930 handwashing test method. The detergent used was 50 times diluted antiviral Wide Haiter EX Power (Kao, Tokyo, Japan). The washing machine test was conducted with a water temperature of 25 °C, as called for by the JIS L 1930 C3G home washing-test method for textile products. The drying method was hang-drying.

### 2.6. Habutae Mask-Wearing Trial Test

This mask was a willow-leaf type that covers the nose and chin area. We asked the participants to choose L or M or S to best fit their face size. For those who still did not have the right size, we asked them to use the rubber stopper to adjust the length of the mask’s ear strap. We asked 100 people to wear these masks for at least 1 h per day in their daily lives for 14 days. The mask-wearer test was performed on 100 subjects. We think the sample size was sufficient. The frequency of washing was left up to the individual. Figure 3 shows information on the subjects. We surveyed 100 people and received responses from 95 subjects; 39 males and 56 females completed the experiment.

Their ages ranged from 20s to 50s, but most were in their 20s. As shown in Figure 3b, 12 people with skin allergies were included in the study. Other allergies included pollinosis and food allergies. In Figure 3c, when asked which kinds of masks they usually wore, 70 people responded that they wore nonwoven masks while 31 said they wore cloth masks.

**Figure 3 materials-14-07391-f003:**
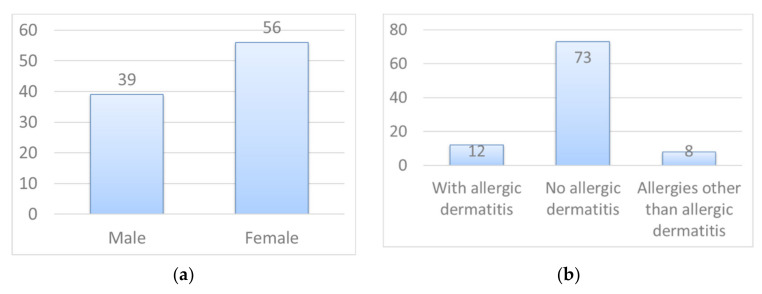
Information on the test subjects. (**a**) Gender; (**b**) allergies; (**c**) types of masks usual used (duplicatable).

## 3. Results

### 3.1. Weave Structure of Silk Fabric and Particle Removal Performance

Figure 4 shows the relationships between the number of sheets and the pressure drop when three types of fabrics with different weave structures (habutae, flat crepe, and twill) were layered. As the number of weave layers increased, the pressure loss increased linearly for all weave structures.

Figure 5 shows the relationship between particle size and removal rate when the number of layers of habutae, flat crepe and twill was increased from one to four. As the number of layers increased, the particle removal rate improved for all weave structures. Twill and flat crepe had the highest and lowest particle removal rates, respectively. However, when only one layer was used, the removal rate of flat crepe was higher than that of habutae. It was found that 40% of particles or droplets with a particle size of 3 μm were removed if the mask was made with two habutae layers, and 80% of particles with a particle size of 10 μm or larger were removed. However, virus particles are 100 nm (=0.1 μm), and habutae fabric alone will not be prevent the virus particles from passing through the mask, no matter how many layers are used.

### 3.2. Nanofiber Treatment on Silk Fabric Surface

Figure 6 shows the filtration performance of habutae fabrics laminated with 0.5 and 1.0 g/m^2^ nanofibers. Nanofiber layer of 1 g/m^2^ on the habutae fabric resulted in the removal of 94% of 0.3-μm particles (pressure loss 195 Pa). When two habutae layers, each with a nanofiber layer, were combined, more than 99% of 0.3-μm particles were removed (pressure loss 339 Pa). Since it is difficult to remove small particles and viruses by woven fabric alone, laminating a nanofiber layer to woven fabric significantly improves particle removal is important. Also the filter performance of ordinary spunbond nonwoven fabric (25 g/m^2^) was about the same as that of habutae. The wearing trial mask of Figure 2 was made of habutae fabric laminated with a 0.5 g/m^2^-nanofiber filter layer (pressure loss 78 Pa).

### 3.3. Washing Resistance of Nanofiber Filters

Figure 7 shows the changes resulting from repeated handwashing of the nanofiber layer laminated on top of habutae fabric. The image labeled “unwashed” shows that the nanofibers were evenly applied by electrospinning to the entire surface of habutae original fabric. On the other hand, repeated handwashing resulted in peeling and tearing of the nanofibers from the intersections of the warp and weft yarns.

### 3.4. Comfortability of Habutae Mask

Figure 8a–e shows the subjects’ responses regarding the comfortability of the habutae mask vs. nonwoven masks. (a) The habutae mask did not shift on the face as much as nonwoven masks do. This is probably due to the fact that the sizes of the masks (L, M, and S) matched well with different face sizes, and to the fact that the habutae mask was made in a more three-dimensional shape than nonwoven masks. (b) We were surprised by the result that the habutae mask felt less stuffy than the nonwoven mask. This may be attributable to the fact that silk is water-absorbent, nanofibers are hydrophilic and the mask has a three-dimensional structure with space around the mouth. This result shows the comfortability of the habutae mask. (c) Silk is a natural fiber and is not considered to have any odor of its own. Also, although a mixture of alcohol and water was used to make the nanofibers, no residual alcohol was thought to be present. (d) The reason for this may be that the shape of the mask is three-dimensional, so that the space around the mouth allows for easy speaking. (e) It is important to note that the habutae mask is less suffocating than the nonwoven mask. We think the most important reason for this is that the habutae mask is wider at the corner of the mouth and has a bulkier structure that wraps around the nose.

### 3.5. Cosmetic Staining and Washing of Habutae Masks

Figure 9(a)–(c) shows the ease of washing and the staining caused by cosmetics on the masks. (a) Many women were concerned about color transfer of lipstick and cosmetic foundation to the mask. Habutae is a woven fabric and has an uneven surface, and it became clear that if lipstick or powder gets into the gaps, it will be difficult to remove. For this reason, the surface of the fabric needs to be stain-resistant. (b) The subjects reported that the masks were easy to wash by hand and almost no trouble to soak them in Wide Haiter. On the other hand, in the case of heavy soiling, it was necessary to wash the mask with a brush or the like, but since we were concerned about damage to the nanofiber layer, we asked the participants not to do so on such occasion. (c) After washing, the masks dried in the shade quickly enough.

### 3.6. Allergy Symptoms Triggered by Wearing a Mask

As Figure 10 shows, no subject developed an allergic reaction. This is an important result of this study. It has been vaguely said by word of mouth that silk masks are good for the skin, but this result is supporting evidence. This may be related to the facts that silk fibers have a low coefficient of friction [20] the fiber diameter is adequate, they do not irritate the skin and silk is biocompatible.

## 4. Discussion

### 4.1. Pressure Drop in Textile

Numerous studies have been conducted on the performance of masks to prevent the transmission of SARS-CoV-2 [21,22]. The pressure drop of a textile can be converted into air flow resistance. Fujimoto et al. [23] reported that there is a linear relationship between the airflow resistance and the thickness of a fabric if the airflow resistance and the volume fraction of the fabric are taken as the property coefficients of the material. We found that the number of layers of silk fabric and the pressure drop can be described by the very simple linear relationship shown in Equation (1).
(1)P=αL

This is due to the fact that all the fabrics were woven with the same silk yarn. In other words, the pressure loss can be expressed only by the coefficient of α, which is determined by the weave structure. This α is determined by the denier of each warp and weft yarn and the number of yarns per unit width, since all raw yarns are identical. This fact makes it very simple to evaluate the breathability of a silk mask.

The pressure drop of a standard nonwoven mask is 10 Pa [24]. In comparison, silk fabrics are made of long monofilament yarns, and the monofilament itself is a straight fiber with almost no crimp, so the woven fabric is densely packed. By reducing the denier of the woven yarn, the yarn becomes thinner and the number of gaps in the woven fabric increases, but the original texture of the fabric is reduced. Figure 11 shows the relationship between coefficient α and the unit weight of the fabric obtained from the equation of relationship between weave structure and pressure loss shown in Figure 4. This is attributed to the fact that the warp yarns of flat crepe are twisted and, after scouring, the warp yarns shrink to create a little grainy texture, which forms moderate gaps. Therefore, flat crepe is more suitable for masks than habutae.

### 4.2. Evaluation of Pore Size of Fabric

The mask used for the wearing test has a structure consisting of two layers of habutae. In between, there is a nanofiber layer. This section discusses the pore size of the habutae fabric. There are two types of pores in the fabric. One is the large pore at the intersection of the warp and weft yarns, as shown in Figure 1. The distribution of pores in habutae is shown in Figure 12a. The other type of pores are gaps between silk fibers. By differentiating (differencing) the graph in Figure 5, we took advantage of the fact that the areas with large changes in particle capture correspond to a large percentage of pores (Figure 12b). As a result, the pore size between fibers was found to be 2.5 μm. It was also found that as the number of layers increased, the effect of the gap between the yarns, shown in Figure 12a, disappeared and the interfiber effect became more pronounced. This interfiber pore size was found to dominate the particle capture rate of the fabric. This means that it is difficult for habutae to capture particles smaller than 2.5 μm.

### 4.3. Washing Durability of Nanofiber Habutae Mask

The garbage problem and ocean pollution caused by non-woven masks are also becoming more serious. For this reason, habutae masks, which can be used repeatedly and do not cause allergies, are gaining attention. The washing method for silk fabrics is hand washing. Figure 13 shows the change in the particle rejection rate of the habutae fabric laminated with 0.5 g/m^2^ of nanofibers used in the trial mask as a function of the number of times it is washed. It can be seen that as the number of washes increases, the particle rejection rate decreases. Also, the performance decreased significantly even after one washing in the washing machine instead of hand washing. We will discuss the cause of this. For the nanofibers, we used EVOH/PU emulsion as the polymer and water/alcohol as the solvent, considering the safety to the skin. Therefore, as shown in Figure 7, the surfactant in washing caused swelling of the nanofibers. In addition, the nanofibers in the pores at the unevenness and intersection of warp and weft yarns were torn. Therefore, in order to improve the washing durability of the nanofibers, we think it would be better to laminate a nanofiber sheet onto a nonwoven fabric or other material and sandwich it between two habutae layers as a filter. Although the performance of the nanofiber filter layer degrades with washing, it retains much higher performance after 10 washes than the habutae alone.

### 4.4. Development of the Mask with Low Pressure Loss and Excellent 0.3-μm Particle Removal Ability

Konda et al. [13] reported that dense fabric layering provides high filtering performance. From a literature review of numerous cloth masks, Clase et al. [25] reported that cloth layering improves filter performance, but they did not discuss the relationship between this improvement and pressure drop. Shen et al. [26] also reported the usefulness of nanofiber filters in comparison with nonwoven fabrics. However, they did not discuss pressure drop. The JIS T9001 standard for masks for general household use in Japan requires that the pressure loss of the mask be less than 60 Pa and that the rejection rate of polystyrene particles of 0.1 μm or 0.3 μm be at least 95% (PFE test). In Figure 14, we plot the results for various masks studied by Teesing et al. [27] together with our results. This comparison shows that it is impossible for a cloth mask alone to comply with the JIST9001 PFE test. Habutae exhibits 17 Pa pressure loss, so wearing trial mask consists of a fabric without a nanofiber layer on the front side and the habutae with a nanofiber layer on one side of the back. The pressure drop of the trial mask is 95 Pa (17 Pa + 78 Pa).

Woven fabrics have a higher pressure drop than nonwoven fabrics. In order to lower the pressure loss, the fiber diameter must be made thinner, but the thickness of silk fiber is 10μm, and it is not possible to make it smaller. Therefore, for silk masks, it is necessary to change the weave from habutae to chiffon to improve air permeability, and to layer nanofibers with excellent washing durability on the surface of the nonwoven fabric and sandwich them between the fabrics as filters.

Despite living in the area where habutae fabric is produced, the younger generation in Fukui Prefecture rarely wears silk products, including habutae, and has the impression that they are troublesome to handle. The habutae face mask in our study provided a good opportunity for the younger generation to rediscover the appeal of silk and to experience the ease of washing it. I hope that silk fabrics will become more and more popular in the future.

## 5. Conclusions

The global spread of SARS-CoV-2 caused a shortage of nonwoven masks. As a result, cloth masks spread rapidly to market. Among cloth masks, those made of silk fabric, which are less likely to cause skin damage and allergic symptoms, have attracted a great deal of attention.

We investigated the filter characteristics of silk fabrics with different weave structures (habutae, flat crepe and twill). The results showed that woven fabrics alone could not sufficiently block particles finer than 1 μm, even when the layers were overlapped.

Therefore, we had a nanofiber filter layer fabricated on the surface of habutae fabric by an electrospinning method at a weight of 1 g/m^2^. The nanofibers were able to remove more than 94% of the 0.3-μm particles, which are similar to the size of virus particles. However, the nanofiber layer was so dense that it caused an increase in pressure drop, so we made a thinner nanofiber layer and fabricated the filter on the surface of the habutae fabric at 0.5 g/m^2^.

A three-dimensional mask consisting of two woven fabric layers, one with a nanofiber layer on the inside and the other with a normal woven fabric without a nanofiber layer on the outside, was fabricated and tested on 95 people. Those subjects judged the nanofiber habutae masks to be more comfortable than nonwoven masks. Moreover, the silk woven masks did not cause allergic symptoms such as skin irritation.

## Figures and Tables

**Figure 1 materials-14-07391-f001:**
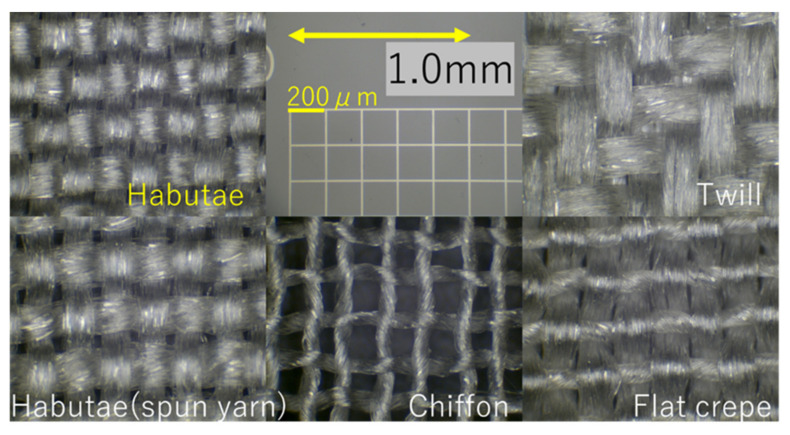
Electron micrographs of typical silk fabrics.

**Figure 2 materials-14-07391-f002:**
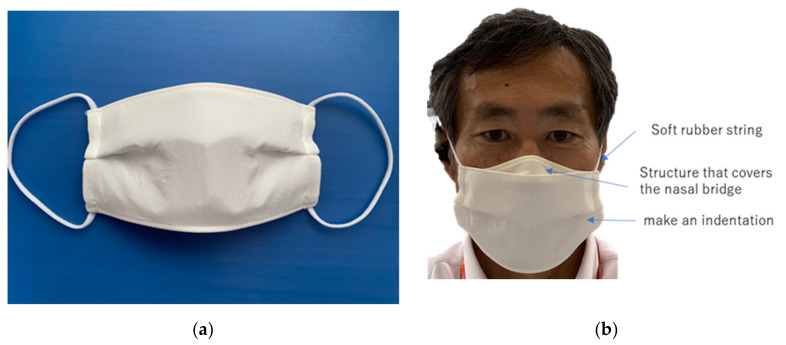
Silk habutae mask with nanofiber filter. (**a**) Overview of the habutae mask. (**b**) Features of habutae mask.

**Figure 4 materials-14-07391-f004:**
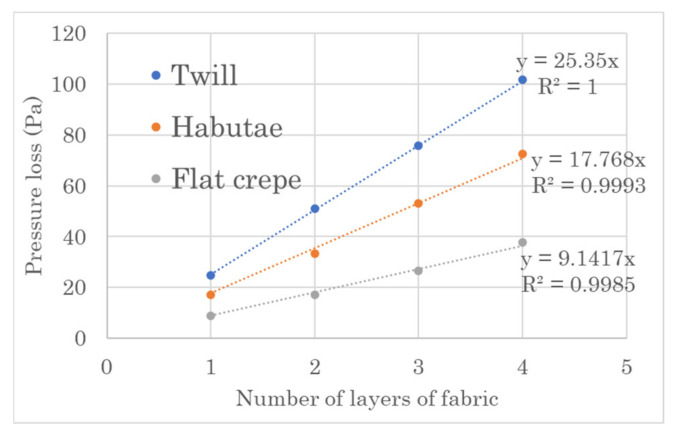
Relationship between the number of layers of three types of silk fabrics (habutae, flat crepe and twill) and pressure loss.

**Figure 5 materials-14-07391-f005:**
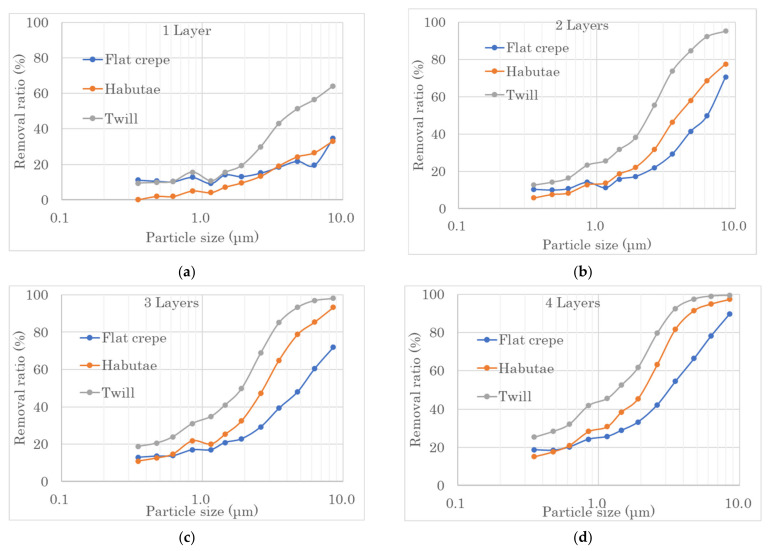
Relationship between the number of fabric layers and the filtration performance of fine particles. (**a**) Filtration performance of fine particles in a single piece of fabric. (**b**) Filtration performance of fine particles in two fabrics. (**c**) Filtration performance of fine particles in three fabrics. (**d**) Filtration performance of fine particles in four fabrics.

**Figure 6 materials-14-07391-f006:**
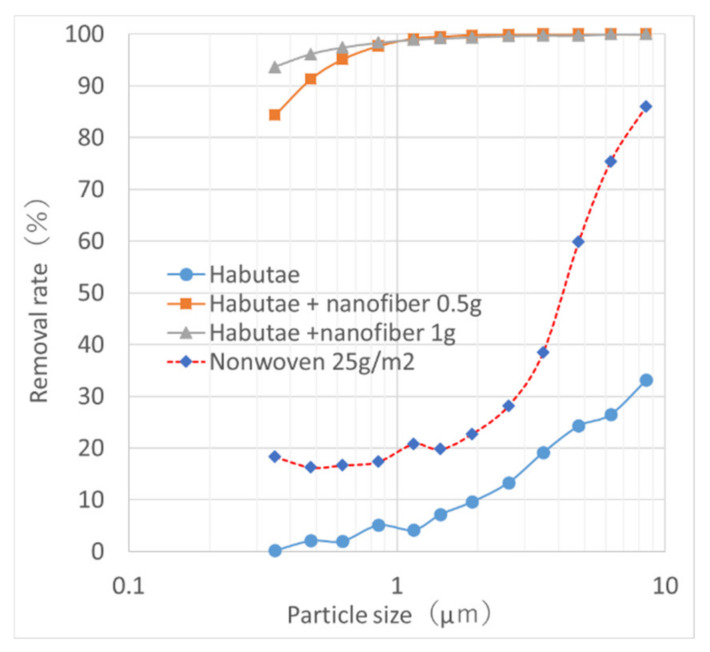
Comparison of filter properties of habutae fabric laminated with nanofibers at weights of 0.5 and 1.0 g/m^2^ and of the fabric without lamination.

**Figure 7 materials-14-07391-f007:**
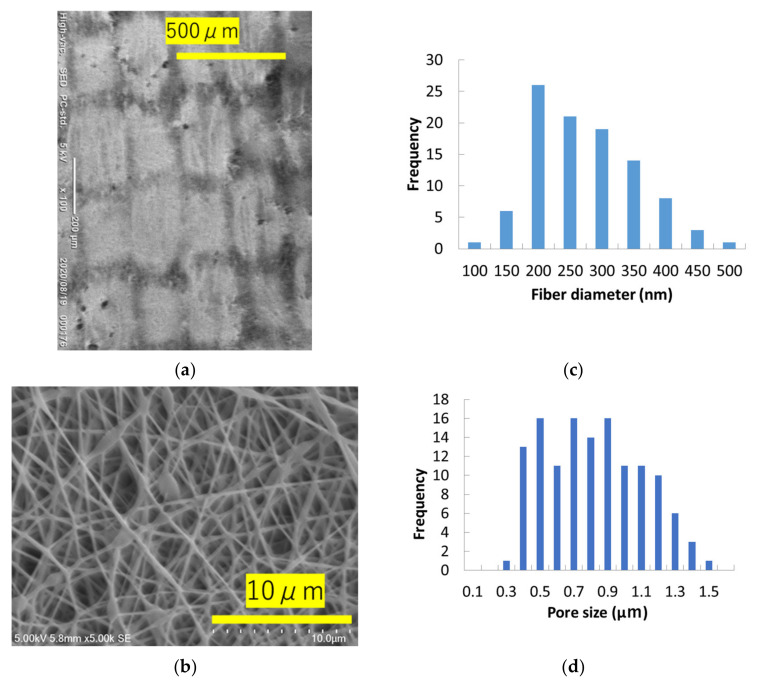
Change in habutae/nanofiber layer before and after washing. (**a**) Nanofibers laminated on habutae (unwashed). (**b**) SEM image of nanofibers (unwashed). (**c**) Fiber diameter distribution of nanofibers (unwashed), average 250 nm. (**d**) Pore size distribution of nanofiber layer (unwashed, average 0.78 μm). (**e**) Nanofibers laminated on habutae (washed five times). (**f**) SEM image of nanofibers (washed five times). (**g**) Fiber diameter distribution of nanofibers (washed five times, average 308 nm). (**h**) Pore size distribution of nanofiber layer (washed five times, average 0.79 μm). (**i**) Nanofibers laminated on habutae (washed 10 times). (**j**) SEM image of nanofibers (washed 10 times). (**k**) Fiber diameter distribution of nanofibers (washed 10 times, average 265 nm). (**l**) Pore size distribution of nanofiber layer (washed 10 times =, average 1.0 μm).

**Figure 8 materials-14-07391-f008:**
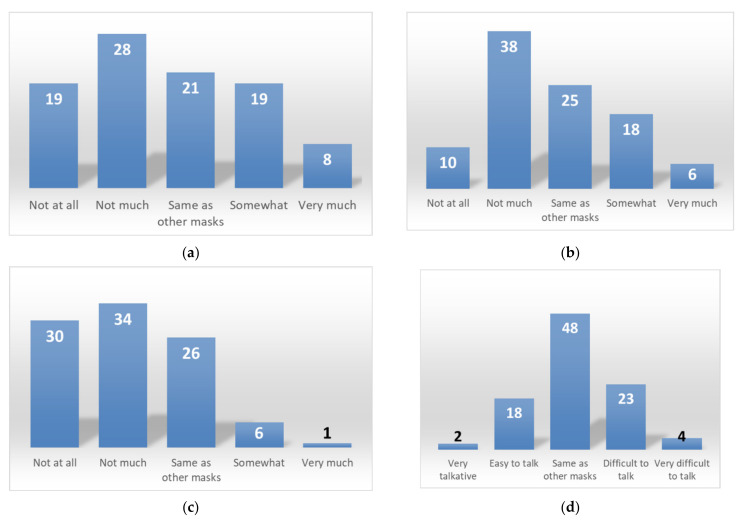
Comfortability of habutae mask: (**a**) shifting during wear; (**b**) steamy feeling while wearing; (**c**) smell while wearing; (**d**) ease of speaking; (**e**) ease of breathing.

**Figure 9 materials-14-07391-f009:**
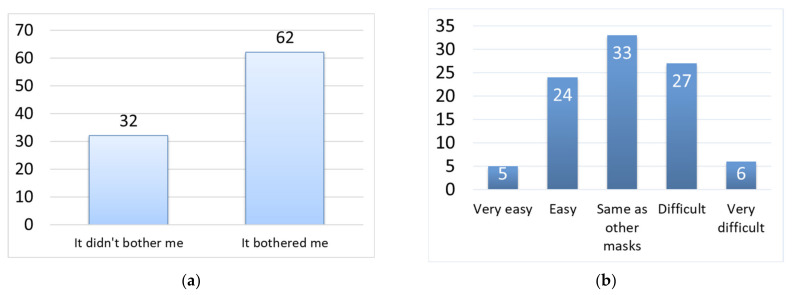
Cosmetic staining and washing of habutae mask: (**a**) cosmetic staining on the mask; (**b**) ease of washing; (**c**) ease of drying.

**Figure 10 materials-14-07391-f010:**
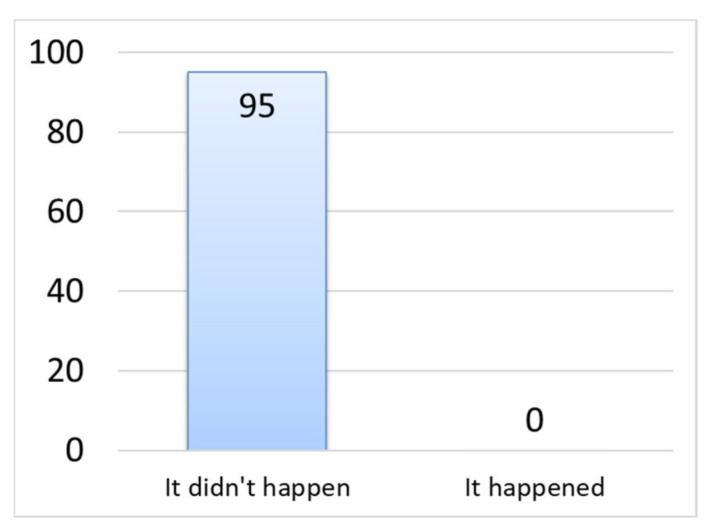
Occurrence of allergic reaction.

**Figure 11 materials-14-07391-f011:**
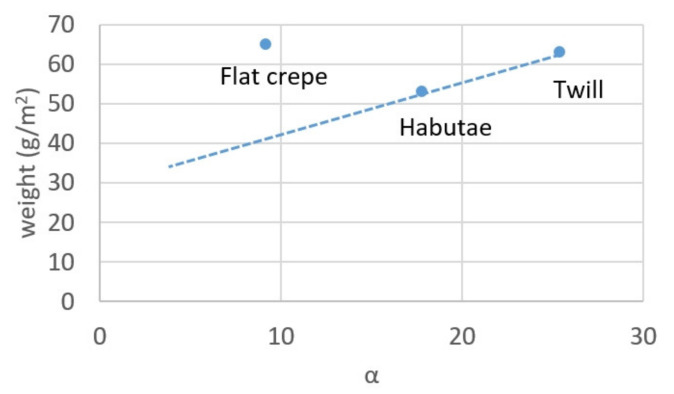
Relationship between coefficient α and unit weight of silk fabrics.

**Figure 12 materials-14-07391-f012:**
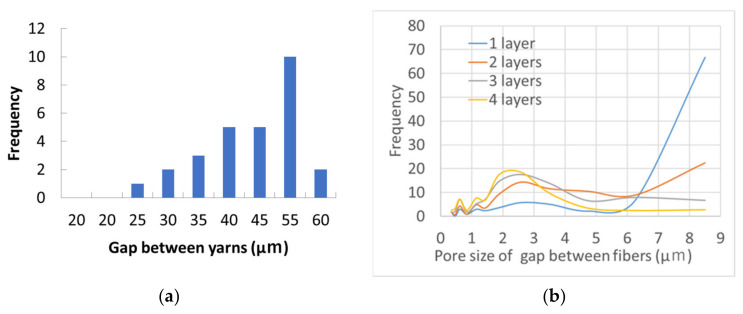
Pore size distribution of habutae fabric. (**a**) Distribution of large pore size formed at the intersection of warp and weft yarns. (**b**) Distribution of pore size between fibers when the number of fabric layers is changed.

**Figure 13 materials-14-07391-f013:**
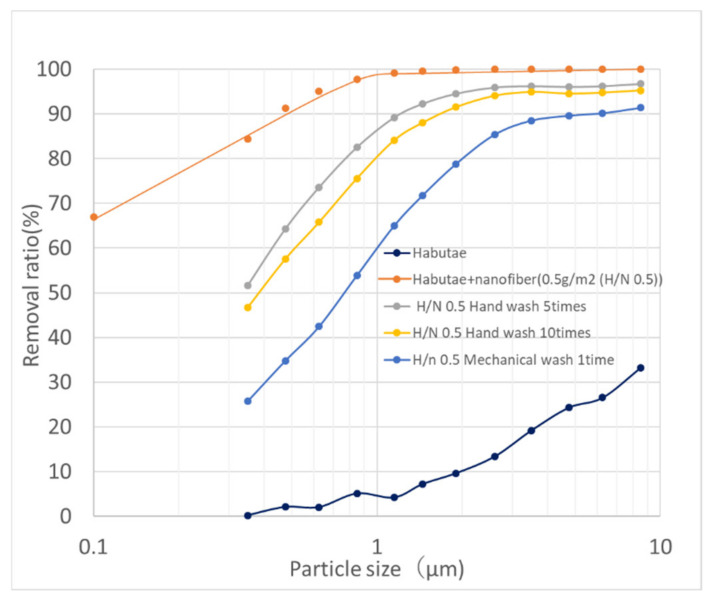
Degradation of filter performance of habutae fabric with electrospun nanofibers (0.5 g/m^2^) by washing.

**Figure 14 materials-14-07391-f014:**
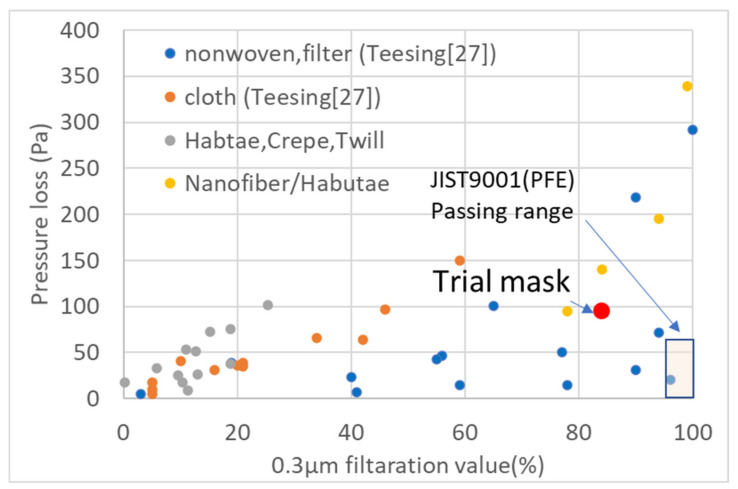
Relationship between 0.3-µm particle-removal rate and pressure drop for masks made from various materials (data taken from Teesing et al. [27]).

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
