# Peer review of "Infection Prevention Mask Consisting of Nanofiber Filter and Habutae Silk Fabrics"

_materials, 2021, doi:10.3390/ma14237391_

Round 1

Reviewer 1 Report

The article provides the use of laminating nanofibers on habutae silk fabric in face masks to prevent or reduce skin irritation and allergic symptoms caused by long-term mask use. Comments for the authors are required to be addressed to improve the manuscript. Please see the attached file.

Author Response

The article provides the use of laminating nanofibers on habutae silk fabric in face masks to prevent or reduce skin irritation and allergic symptoms caused by long-term mask use. I have some comments for the authors: required to be addressed to improve the manuscript.

  1. The manuscript did not cover the present state-of-the-art electrospun membranes for face masks.

However, some relevant review papers are provided below;

  1. Face Masks and Respirators in the fight against the COVID 19 pandemic: A review of current materials advances and future perspectives.
  2. Jeremy Howard, Austin Huang, Zhiyuan Li, Zeynep Tufekci, Vladimir Zdimal, An evidence review of face masks against Covid 19. 

The references were cited as 22 and 23 and added to the description of the mask.

  1. The characterization methods of the reported article are not explained clearly. For example, Figure1 shows the electron micrographs of silk fabrics; however, it does not mention how the images were collected.

The explanation of Figure 1 was missing from the text, so it has been added. Line97

  1. The article does not give the average pore sizes for the different silk fabrics and the synthesized nanofibers.

Since the pore diameter/size varies throughout the nanofiber, the normal distribution of pore diameter/size of the control, different silk fabric, and nanofibers will give a better understanding of the fibers and their application in facemask. Please include SEM analysis. This is also important to analyze the relationship between particle size and removal rate when the number of layers was increased. 

The fiber diameter of the nanofibers and the pore diameter of the nanofibers were added to Figure 4. The pore diameter of the fabric is added in Figure 13.

  1. Which material was considered as the control? On page 14, line 173 mentions habutae fabric laminated with 1g/m2 nanofibers has pressure drop 195 Pa. How many layers of fabrics are used here? It seems the pressure drop is high compared to the nonwoven mask reported by the author as 10 Pa.

Higher filtering performance results in higher pressure drop.

By nature, nanofibers are supposed to be resistant to high pressure loss, but due to the random direction of nanofibers, it is not easy to achieve low pressure loss.

In this research, the amount of coating is controlled by the weight per square meter of nanofiber.

This coating amount is adjusted by the speed of the substrate passing through the nozzle.

When the amount of nanofibers applied was 1g, it was 195Pa, which is very high.

When the amount of nanofiber applied was reduced to 0.5g, the pressure loss was reduced to 78Pa.

This was used as the filter for the wearing test mask.

A single feather layer is 17 Pa, so this mask consists of a fabric without a nanofiber layer on the front side and a feather layer with a nanofiber layer on one side of the back.

The pressure drop of the entire mask is 95Pa (17Pa + 78Pa). Line 336-338

10Pa is the ultimate target value, and the acceptable range is 60Pa. We could not reach it this time, but we discussed the method to achieve it in the discussion.  Line345-348

  1. The article reports a correlation between pressure drop and the number of handwashing of the mask. A graphical presentation would give a clear understanding.

We have added SEM pictures of the number of washes and fiber shape for each in Figure 7 and added explanations.

  1. There is no data to justify that habutae mask does not shift on the face as much as nonwoven masks do. Please explain.

The mask is a willow leaf type that covers the nose and chin area. There were three sizes to choose from: L, M, and S. We asked the participants to choose the one that best fit their face size. For those who still did not have the right size, we asked them to use the rubber stopper to adjust the length of the mask's ear strap. I have added these. Line144-147

  1. The controllability and repeatability of the data are not reported here. For example, how many tests were done on each fabric? What was the % of error involved? 

Pressure drop and particle rejection were measured at N = 3. The fiber diameter and pore size of the nanofibers were also measured in three images. All of the data in the middle of the three measurements were used for publication.

The variability between each measurement was all within ±15%. This has been added.Line111-113

  1. Figure 12 does not specify which last three curves(Hand washing five times, Handwashing 10 times, and Mechanical washing one time) belong to Habutae or Habutae with nanofiber fabric.

Figure 12 shows the case where 0.5g of nanofibers are laminated onto a sheet of habutae. The explanation has been added. This figrure12 changes figure13.

  1.  

The typo has been corrected.

  1. Captions on page 8 are not consistently labeled. For example, figure 8d has a caption below the chart; meanwhile, the other does not.

The captions in the figures have been unified.

The text and legend of some figures are not clear; for example, Figure 6, Figure 9. 

The problems with the legend and other figures have been fixed. The Habutae + nanofiber 0.5g in Figure 12 is identical to that in Figure 9. The explanation has been added to the explanation of Figure 12.

Reviewer 2 Report

This is an organized and prepared work and can be accepted after careful revision. Below are the comments.

  1. The introduction should be revised to give clear motivation and why the result is important. What are the novelty and major advances of this current work?  
  2. Is the preparation method can be scalable?
  3. The stabilities of the fibers under different conditions are important for commercial applications. How about the long time stabilities of the fibers under different temperatures, moisture contents and pH values?
  4. Are the fibers can be recycled after use?

Author Response

The introduction should be revised to give clear motivation and why the result is important. What are the novelty and major advances of this current work? 

The purpose of this research has been added to the introduction in an easy-to-understand manner. Line52-88

Is the preparation method can be scalable?

Habutae fabric is a typical silk fabric. We added it.Line95-99

The wearing test was conducted on 100 people. We think that the number of people is sufficient. Line 144-151

The electrospinning equipment was the smallest general-purpose manufacturing equipment.

For the solution, we selected a highly safe polymer solution.

We added these. Line83-88

The stabilities of the fibers under different conditions are important for commercial applications. How about the long time stabilities of the fibers under different temperatures, moisture contents and pH values?

The chemical stability of silk has been investigated in detail in the literature. References 24 and 25 have been added.

Are the fibers can be recycled after use?

Silk can be recycled. EVOH fiber is not recyclable, but it is a chemically and thermally stable material. I added citations of 24~27. Line52,85-86

Round 2

Reviewer 1 Report

no more comments. 

Reviewer 2 Report

The manuscript has been improved by the authors carefully based on the Reviewers' comments. The current version is  suggested to be accepted as it is.